# A Pilot Study on Emotional Equivalence Between VR and Real Spaces Using EEG and Heart Rate Variability

**DOI:** 10.3390/s25134097

**Published:** 2025-06-30

**Authors:** Takato Kobayashi, Narumon Jadram, Shukuka Ninomiya, Kazuhiro Suzuki, Midori Sugaya

**Affiliations:** 1Graduate School of Engineering and Science, Shibaura Institute of Technology, Tokyo 135-8548, Japan; al20009@shibaura-it.ac.jp (T.K.); nb23107@shibaura-it.ac.jp (N.J.); al20113@shibaura-it.ac.jp (S.N.); 2Nomura Co., Ltd., Tokyo 100-8130, Japan; ka.suzuki@nomura-g.jp

**Keywords:** virtual reality, emotion evaluation, semantic differential method, EEG, heart rate variability

## Abstract

In recent years, the application of virtual reality (VR) for spatial evaluation has gained traction in the fields of architecture and interior design. However, for VR to serve as a viable substitute for real-world environments, it is essential that experiences within VR elicit emotional responses comparable to those evoked by actual spaces. Despite this prerequisite, there remains a paucity of studies that objectively compare and evaluate the emotional responses elicited by VR and real-world environments. Consequently, it is not yet fully understood whether VR can reliably replicate the emotional experiences induced by physical spaces. This study aims to investigate the influence of presentation modality on emotional responses by comparing a VR space and a real-world space with identical designs. The comparison was conducted using both subjective evaluations (Semantic Differential method) and physiological indices (electroencephalography and heart rate variability). The results indicated that the real-world environment was associated with impressions of comfort and preference, whereas the VR environment evoked impressions characterized by heightened arousal. Additionally, elevated beta wave activity and increased beta/alpha ratios were observed in the VR condition, suggesting a state of high arousal, as further supported by positioning on the Emotion Map. Moreover, analysis of pNN50 revealed a transient increase in parasympathetic nervous activity during the VR experience. This study is positioned as a pilot investigation to explore physiological and emotional differences between VR and real spaces.

## 1. Introduction

### 1.1. Background

Virtual reality (VR) is a technology that enables users to experience three-dimensional computer-generated environments as if they were physically present within them [1,2]. Because VR allows users to engage with highly realistic simulated environments [3,4], it has been increasingly applied in various fields where providing real-world experiences is difficult or costly, offering a convenient and efficient alternative [5,6,7].

One such field is architecture. In architectural design, allowing users to experience a space before construction typically incurs high development costs. By enabling users to explore a designed space within a VR environment, it becomes possible to reduce construction costs and evaluate various spatial design options. As a result, the application of VR in this domain is expected to expand. In particular, within interior design—a subfield of architecture—designers consider aesthetics, functionality [8,9], psychological effects, lighting, cost, and other factors to create optimal environments for users [10,11,12,13,14,15]. In recent years, the use of VR environments to support the evaluation of such design proposals has gained attention [16,17,18,19]. Because interior design often involves complex combinations of elements tailored to diverse user preferences, recreating and evaluating all possible variations in real space is impractical. VR offers a promising solution by enabling the virtual reproduction of spatial designs on a computer, allowing users to experience various design options and compare them to those in real space [16,17,18,19].

In VR spaces, elements such as furniture and wall colors can be easily modified by changing parameters, making it easier to examine alternative design configurations [20,21]. Dorta et al. note that VR is a useful tool for evaluating spatial designs, identifying potential design issues in advance, and reducing time and material costs [22]. Moreover, comparisons conducted in VR do not require the physical construction and subsequent disposal of materials, offering a sustainable and efficient alternative for evaluating design options.

Meanwhile, psychological evaluation of the user’s experience in a given space has been emphasized as an important aspect of spatial design [18,23]. For example, a study by Xiang et al. demonstrated that specific office designs can contribute to stress reduction and improved comfort for users [18]. Similarly, Robert et al. reported a strong relationship between psychological stress, comfort, and emotional state [23]. These findings suggest that spatial design is closely associated with the emotional responses of users, and that designers should take these emotional factors into account.

As VR technology continues to advance, it is conceivable that designers will increasingly develop and present design proposals within VR environments. In such scenarios, it would be possible for designers to evaluate users’ emotional responses in VR before implementing the designs in real space. If VR-based evaluation proves reliable, users may be able to select a final design based solely on their experiences in the virtual environment, without the need to experience the actual constructed space. This would offer significant advantages in terms of reducing pre-construction development costs.

However, to support this approach, it is essential to verify that emotional experiences elicited in VR spaces are consistent with those experienced in real spaces. Previous studies have shown mixed results regarding this equivalence. For example, Llinares et al. [24] reported minimal cognitive differences, whereas Marín-Morales et al. [25] observed physiological divergence between conditions.

Despite such findings, the emotional equivalence between VR and real spaces has not been sufficiently established. For instance, if a user perceives a VR space as comfortable, it must be assumed that the same design would evoke a similar response in real space. In particular, few studies have conducted rigorous comparisons between VR and real spaces using identical spatial designs, and the emotional responses elicited by each have not been thoroughly investigated. Although it is possible that emotional experiences in VR and real spaces differ, this issue has not been sufficiently examined or discussed in previous research.

Given these gaps in the literature, the present study is positioned as a pilot investigation aimed at exploring emotional and physiological differences between VR and real spaces using identically designed environments.

### 1.2. Challenges

Numerous studies have investigated experiences within virtual reality (VR) environments. For example, Li et al. evaluated different VR spaces and revealed that higher arousal states enhance cognitive control, which refers to the brain’s ability to regulate thought and behavior [26]. Similarly, Kühne et al. compared VR videos presented through a head-mounted display (HMD) with videos shown on a standard monitor, demonstrating that VR induces greater immersion and emotional effects, along with generally heightened physiological arousal [27]. Such findings suggest that VR can provide realistic experiences that evoke physiological responses similar to those occurring in the real world.

However, these studies primarily focus on comparisons between different VR environments or virtual spaces, and do not compare experiences in VR and real spaces. If the difference in physiological arousal between VR and real space is minimal, it would support the hypothesis that VR can substitute for real spaces. Nevertheless, such hypotheses have not been sufficiently verified.

One notable study comparing the same spatial design in VR and real space was conducted by Llinares et al. [24]. They found no significant differences in cognitive and physiological responses, suggesting that VR could serve as a viable evaluation method. However, their investigation focused only on cognitive aspects such as attention and memory, without assessing emotional differences.

Another study by Marín-Morales et al. evaluated autonomic nervous responses using heart rate variability when participants experienced identically designed VR and real spaces [25]. Their results indicated distinct autonomic responses between the two conditions. However, their experiment lacked a clearly defined physiological baseline, making it difficult to accurately evaluate changes due to the stimuli. Furthermore, their emotional evaluation focused solely on arousal and did not consider other emotional dimensions, such as valence.

Additionally, in both Llinares’s and Marín-Morales’s experiments, although visual stimuli were matched between VR and real conditions, auditory and olfactory stimuli differed. As a result, these studies do not offer a strict comparison based solely on visual input. Therefore, there remains a need for research that strictly controls non-visual environmental factors, such as sound and scent, and compares VR and real spaces under conditions where only visual stimuli differ.

To objectively evaluate emotional responses to spatial design, it is essential to apply quantitative measures of emotion. Among widely used methods, the Semantic Differential (SD) method has been used to evaluate impressions by presenting participants with bipolar adjective pairs [28]. The SD method is particularly effective in assessing emotional impressions. However, it has not been sufficiently applied to evaluate emotional differences between VR and real spaces.

Furthermore, emotional responses experienced in spatial environments are not always verbalizable. Unconscious emotional changes are also significant. In recent years, methods using physiological responses have been proposed to quantify such unconscious changes [29]. These approaches offer the advantage of objectivity, being unaffected by memory, bias, or prior knowledge [30,31].

In particular, physiological indices such as electroencephalography (EEG) and heart rate variability (HRV) allow for real-time tracking of unconscious emotional responses and are unaffected by subjective factors [30,31]. Moreover, their reliability and accuracy in emotional evaluation have been validated in previous research [31], making them well-suited for clarifying differences in emotional experiences between identically designed VR and real spaces. Nevertheless, to our knowledge, few studies have applied such physiological methods to evaluate emotional differences between VR and real spaces, and discussions on which physiological indices to use and what they reveal remain insufficient.

## 2. Proposal

### 2.1. Objective and Methodology

As discussed in the previous section, in the context of space design, it is important to examine whether VR-based spatial experiences can serve as a substitute for experiencing designs in a real space, particularly given constraints such as cost and environmental impact. However, there remains a challenge in that the emotional responses elicited by VR spaces and real spaces have not been sufficiently evaluated or discussed under controlled conditions. Moreover, there is a lack of studies that objectively assess emotional experiences using appropriate evaluation methods, especially in comparing VR and real spaces designed with identical spatial layouts. Given the lack of prior research directly comparing emotional responses between VR and real spaces, this study aims to objectively evaluate the emotional responses elicited by experiencing identically designed VR and real spaces, and to clarify the differences between them using objective evaluation methods.

To achieve this, we constructed a pair of identically designed VR and real spaces and conducted an experiment to evaluate emotional responses. We carefully designed the pilot experiments by preparing the pairing of the space and the evaluation method. Through these analyses, we aim to investigate and discuss the emotional differences between VR and real spaces, as highlighted in the challenges described earlier. This study is positioned as a pilot investigation to explore potential differences using physiological and subjective indices.

We considered the following methodological approach:

(1) We constructed paired environments in VR and the real world, maintaining consistent spatial design across both. For two distinct space types, VR environments were primarily developed to allow direct comparison with corresponding real-world spaces. Further details are provided in Section 2.2. We executed a pilot experiment using the environment. For the evaluation methodology, we consider the following.

(2) Evaluations were conducted to assess emotional responses elicited in both environments. To ensure objectivity, we utilized a combination of cognitive and physiological measures. For the subjective emotional responses, we evaluated using a cognitive impression method, namely the Semantic Differential (SD) method. For emotional responses that are not subjectively recognized, and/or not always fully expressible through self-report, we employed physiological indices such as electroencephalography (EEG) and heart rate variability. The details are described in Section 2.3.

(3) The analysis method is described in Section 2.4. The analysis method includes not only the statistical approaches, but also the visualization techniques (Emotion Map) to aid in our understanding of the emotional evaluation and the emotional reactions more comprehensively.

Consequently, this study is positioned as a pilot investigation exploring the feasibility of using multimodal indicators to detect emotional differences between VR and real-world experiences. Through these analyses, we aim to elucidate the emotional distinctions between virtual and physical environments, thereby addressing the central issue identified at the outset of this research.

### 2.2. The Pair of Identically Designed VR and Real Spaces

To achieve the objectives described above, we developed spatial environments with identical designs in both VR and real-space formats. Two types of spatial designs were prepared (Figure 1), based on the assumption that each would elicit different types of emotional responses.

Figure 1 shows the environments referred to as Place 1 and Place 2. Place 1 was designed as a conference space intended to promote concentration and creativity. The space features natural lighting, a large depth (approximately 39 m × 16 m), and the scent of cypress, all contributing to an environment that supports focused work.

In contrast, Place 2 (Figure 1) was designed as a reset space with the purpose of promoting relaxation. The concept behind the reset space is to provide users with a momentary emotional reset, allowing them to step away from work. This space does not include strong scents or expansive depth; rather, it features a relatively narrow visual field with the presence of potted plants. For both types of environments, VR versions were developed to serve as comparison targets with the real space.

### 2.3. Evaluation Method

This study aims to compare impressions, emotional responses, visual load, and memory engagement when viewing virtual reality (VR) and real spaces, and to discuss the differences between them. Impressions were assessed using subjective evaluation based on a questionnaire following the Semantic Differential (SD) method.

For emotion estimation, various approaches have been proposed in prior research [26,27,30,31,32,33]. In this study, physiological reactions such as brainwaves and heart rate were measured to estimate emotional states, based on the advantages discussed in Section 1.2. Unlike facial expressions or voice, physiological responses cannot be consciously controlled by individuals, making them suitable for objectively evaluating reactions to stimuli [30]. Thus, this method was considered appropriate for the present experiment.

Previous research has also demonstrated that brainwaves can be used to evaluate visual load during VR use [34,35] and cognitive load associated with memory processing [36,37]. Based on these findings, this study adopted a method using electroencephalogram (EEG) and heart rate measurements to assess emotional responses during the experience.

Through the analysis of physiological data, differences in visual load and memory-related activity will be examined statistically. Since physiological indices can reveal subtle differences in responses to stimuli, they allow for the evaluation of emotional states without interfering with the user’s real-time spatial experience.

The following sections describe the questionnaire and physiological indices used in this study, as well as the method for evaluating emotional responses.

#### 2.3.1. Cognitive Evaluation: Semantic Differential (SD)

Based on previous studies, a questionnaire using the Semantic Differential (SD) method was employed to assess participants’ impressions of the spaces. The SD method evaluates a target using multiple pairs of opposing adjectives [28]. This allowed for a comparison of impressions evoked by viewing real and VR spaces.

Referring to earlier studies that evaluated impressions and emotions related to spatial elements such as materials [38], lighting [39], and form [40], ten adjective pairs were selected from those previously used (Table 1) [28,38,39,40,41]. These pairs were chosen to comprehensively capture the sensory characteristics of the space. Because these evaluations are based on intuitive impressions, they are considered suitable for comparing how real and VR spaces influence perception.

Each adjective pair was rated on a 7-point Likert scale, with participants asked to respond based on their immediate impression of the most recently viewed space, without overthinking their answers.

#### 2.3.2. Electroencephalography (EEG)

EEG-based assessment has already been proposed in previous research on VR environments for the evaluation of emotional responses in VR and real spaces using physiological indices. Kühne et al. demonstrated that when the same video was presented in VR and on a 2D display, the VR condition induced greater immersion, stronger emotional reactions, and heightened arousal compared to the 2D condition [25]. Similar trends have also been reported by Tian et al. [42]. Based on these findings, this study uses EEG-based indices to evaluate arousal levels.

Furthermore, multiple EEG indices are used to provide a comprehensive assessment. It is known that emotionally impactful events are more likely to be remembered [43], and that impressions formed during experiences can influence emotional responses [44]. Therefore, if the impressions formed while viewing VR and real spaces differ, it is reasonable to expect corresponding differences in emotional response and memorability.

In addition, prior studies have reported that VR use can impose visual strain and lead to perceptual fatigue distinct from experiences in real space [45]. Such fatigue may provoke negative emotions, thereby influencing the emotional perception of a space [46]. Previous studies have also shown that EEG can be used to evaluate visual load [34,35] and memory-related activity [36,37], supporting its appropriateness as a physiological measure for this study.

Based on these findings, appropriate EEG indices were selected from the measured signals. EEG reflects electrical activity in the brain and can be classified into several frequency bands based on signal characteristics [47]. Alpha waves are commonly associated with relaxation and restful states [48,49], while beta waves are linked to attention, cognitive activity, stress, and arousal, with their power increasing during heightened arousal [48,49].

In this study, the beta/alpha (β/α) ratio was used as an indicator of arousal. A higher β/α ratio suggests a state dominated by beta waves, indicating increased attention, stress, and arousal. In contrast, a lower β/α ratio reflects a state in which alpha waves are more prominent, suggesting a relaxed or restful state with lower arousal.

Theta waves are associated with memory encoding, retrieval, and the maintenance of working memory [34,49,50]. In addition, based on prior studies [34,36], gamma waves were evaluated to investigate the effects of sensory processing and cognitive load. Gamma waves were categorized into low gamma (30–50 Hz) and high gamma (50–70 Hz). Although some previous studies have used 50–100 Hz to define high gamma, this study limited the upper bound to 70 Hz to reduce the influence of artifacts.

Low gamma activity is often associated with the integration of sensory input—such as visual, auditory, or somatosensory stimuli—and is known to increase under high perceptual load or when attention is focused on a specific stimulus [36,37]. It is also believed to reflect localized synchronization of neural networks and increases during tasks involving high short-term cognitive load [34,35,50].

In contrast, high gamma activity is involved in higher-order cognitive processes such as memory encoding, retrieval, and visual information integration [34,35,50]. It has been reported to increase during episodic memory retrieval [45,46,49] and visual information processing [36,37]. Additionally, interactions between high gamma and theta waves have been linked to successful memory recall [34,35,50].

Based on these findings, this study uses theta waves and low gamma and high gamma activity to investigate differences in short-term memory, memory encoding and retrieval, and visual load when viewing spaces. The selected EEG indices and their interpretations are summarized in Table 2.

The human brain relies on the frontal lobe for intellectual functions such as reasoning, judgment, and creativity, and this region is commonly used in emotion measurement. Based on these considerations, this study measured EEG signals at the AF4 electrode site, which corresponds to the frontal lobe, in accordance with the international 10–10 system [51].

EEG signals obtained from the EEG device may contain noise due to various factors, including body movements. To ensure accurate analysis, artifact removal was performed. Specifically, outlier correction was applied by calculating the standard deviation (σ) of the EEG data and adjusting values that exceeded ±3σ.

#### 2.3.3. Heart Rate Variability (HRV)

In this study, heart rate variability (HRV) indices were used to capture physiological responses during experiences in virtual reality (VR) and real spaces, as indicators of emotional states. HRV indices quantify fluctuations in heart rate over time. Among various HRV measures, we employed pNN50, a primary time-domain metric for quantifying HRV [52]. pNN50 enables emotion-related HRV analysis within very short time intervals [52], and it is known to be strongly associated with parasympathetic nervous activity [52]. Higher pNN50 values are interpreted as reflecting emotionally relaxed states with lower psychological stress [52,53], whereas lower values suggest higher psychological tension or stress [52,53]. Emotionally relaxed states are generally categorized as positive emotions, and psychologically tense states as negative emotions. Based on these associations, this study used the time-series variation in pNN50 to evaluate emotional changes during the spatial experiences.

pNN50 is defined as the percentage of successive heartbeat intervals that differ by more than 50 ms. The formula for calculating pNN50 is shown below. Here, NN50 represents the number of interval pairs with differences exceeding 50 ms, and N indicates the total number of heartbeat intervals:$$pNN50=\left(\frac{NN50}{N-1}\right)\times 100$$

### 2.4. Estimation and Visualization of Emotion Using EEG and HRV

To compare emotional responses elicited during the spatial experiences, this study employed the Emotion Map [54], a method that estimates and visualizes emotional states based on EEG and HRV data. The Emotion Map visualizes emotional states by mapping them onto Russell’s circumplex model of affect, which is widely used in psychology. This model represents two dimensions of emotion: the vertical axis corresponds to arousal (aroused to drowsy), and the horizontal axis to valence (pleasant to unpleasant) [55].

In the Emotion Map, valence is estimated using HRV indices, and arousal is estimated using EEG indices. These values are plotted as coordinate positions in a two-dimensional space corresponding to Russell’s model, enabling visualization of the emotional impact of each stimulus. The map is divided into four quadrants: the first quadrant represents high arousal and high valence, the second quadrant high arousal and low valence, the third quadrant low arousal and low valence, and the fourth quadrant low arousal and high valence.

Using this method, emotional responses during VR and real-space experiences were compared based on physiological signals. The β/α ratio was used as the EEG-based arousal index (Y-axis), and pNN50 was used as the HRV-based valence index (X-axis). The structure of the Emotion Map is illustrated in Figure 2.

As emotional responses vary between individuals [56], the EEG and HRV indices were normalized for each participant using a robust z-score. This method standardizes data using the median and median absolute deviation (MAD), and is less sensitive to outliers compared to standard deviation-based methods. In this study, no scaling coefficient (e.g., 1.4826) was applied, and the focus was placed on relative comparisons across individuals.

Because the origin of the Emotion Map reflects a baseline emotional state, it was defined based on physiological indices recorded during a resting state. For consistency, the resting state in VR evaluations was also conducted in the VR space. As a result, both the resting and stimulus phases involved wearing a head-mounted display (HMD), thus eliminating the influence of the device itself and allowing emotional responses to be attributed solely to the VR stimulus. To accurately evaluate stimulus-induced emotional changes, it is critical to establish a reliable baseline relative to the resting state. In this study, a regression-based method was employed to address this issue. Over 90% of participants reported having no or minimal prior experience with VR. As physiological responses are known to be influenced by familiarity with the stimulus (a phenomenon known as habituation) [57], unfamiliar stimuli tend to evoke stronger physiological reactions.

To minimize the influence of habituation, a regression-based baseline estimation method was applied. This approach involved performing a regression analysis on the physiological indices recorded during the 2 min resting phase to predict their values over the subsequent 2 min period in the absence of any stimulus. The predicted values were then used as the baseline for emotional evaluation. By modeling the time-series trend of physiological indices and extrapolating beyond the resting phase, this method accounts for individual variability and temporal drift, contributing to more reliable baseline estimation.

## 3. Experiment Method

### 3.1. Overview

The objective of this experiment was to evaluate physiological and psychological responses when participants viewed a real space and a corresponding virtual reality (VR) space that replicated the same design. To achieve this, responses were assessed using physiological indices—specifically EEG and heart rate variability—and a questionnaire based on the Semantic Differential (SD) method. The experiment involved 42 healthy adult participants (29 males and 13 females), with attention given to gender balance.

### 3.2. Equipment and Spaces Used

As shown in Figure 3, this experiment employed a head-mounted display (HMD) for VR presentation and two types of sensors to collect physiological data. The HMD used was the Meta Quest 2 [58], which features a resolution of 1832 × 1920 pixels per eye and a refresh rate of 72 Hz. For EEG measurement, the Ganglion Board by OpenBCI [59] was used, which supports a 4-channel input with a sampling rate of 200 Hz. For electrocardiography (ECG), the MyBeat device by UnionTool [60] was employed. This ECG sensor is attached to the chest using dedicated electrodes and has a sampling rate of 1000 Hz.

### 3.3. Experiment Place

As we described in Section 2.2, we developed a pair of identically designed VR and real spaces. For each set of two spaces, it was necessary to carefully consider the measurement locations. We selected these for both the real space and VR space (Figure 4). Measurement Place 1, corresponding to Place 1 shown in Figure 4, is a space with a large number of furnishings, such as chairs and desks, featuring a significant depth that allows visibility to the back of the room.

Measurement Place 2, used for Place 2, has fewer furnishings compared to Measurement Place 1 and is characterized by visible wooden flooring and plant leaves, resulting in a relatively large proportion of plants in the visual field. These two measurement locations were selected in order to compare and examine results under conditions of different spatial purposes. As noted in Section 1.2 (Challenges), in this experiment, the influence of non-visual factors such as scent and sound was minimized. Therefore, the VR experiences were also conducted at the same physical locations as the real-space measurements (Figure 4). Since participants wore a head-mounted display (HMD) in the real space, the environmental conditions, such as ambient sound, room temperature, and scent, were consistent with those of the real space.

### 3.4. Experimental Procedure

The experimental procedure is outlined below and illustrated in Figure 5:(1)The participant wears the ECG sensor and, while blindfolded to avoid seeing the space, moves to the measurement chair at the designated location.(2)After sitting in a chair surrounded by partitions to prevent visibility of the real space, the participant is fitted with the EEG device. If the VR condition is presented, the HMD is also worn.(3)A two-minute resting period is conducted.(4)The participant views the stimulus space for two minutes.(5)If the HMD was worn, it is removed, and the participant completes the questionnaire.(6)If the VR space was presented in steps (3) and (4), the same procedure is repeated using the real space, and vice versa.(7)The participant is blindfolded and moved to a different measurement location.(8)Steps (2) to (6) are repeated at the new location.

To control for order effects, the presentation order of VR and real spaces and the order of Measurement Places 1 and 2 were randomized. For the resting condition in step (3), partitioned versions of each stimulus space (VR and real) were used to obscure the design features. In the experiment design, the resting phases preceding the VR stimuli (VR1 and VR2) were conducted using the VR space. Participants were instructed to remain still and observe the space calmly, avoiding sudden movements, to prevent signal noise caused by body motion or jaw clenching.

### 3.5. Participants

The participants in this experiment were male and female university students in their twenties, with a total of 42 individuals (29 males and 13 females). All participants had either never used VR before or used it only rarely. The study was conducted with the approval of the Ethics Committee of Shibaura Institute of Technology. Written informed consent was obtained from all participants prior to the experiment.

## 4. Experimental Analysis and Result

### 4.1. Statistical Analysis and Evaluation Overview

As outlined in Section 2, we employed both cognitive and physiological indicators to objectively evaluate emotional responses elicited in VR and real-world environments. The analysis proceeded through the following multi-step methodology:**Subjective Evaluation of Emotional Responses:** Subjective emotional responses were assessed using the Semantic Differential (SD) method, a widely used cognitive impression technique. The results of this analysis are detailed in Section 4.2.**Physiological Evaluation—Brainwaves:** In Section 4.3, we evaluated physiological responses not consciously perceived by participants but elicited by the spatial environments. First, we conducted a basic assessment of alpha and beta brainwave activity. Alpha waves are typically associated with relaxation, whereas beta waves are linked to attention and arousal. To quantify the balance between these states, we computed the β/α ratio, comparing values across the VR and real-world conditions to determine the dominant emotional state.**Memory-Related Brain Activity:** As discussed in Section 2.3.2, theta waves are implicated in memory encoding, retrieval, and working memory maintenance [34,49,50]. We also analyzed gamma wave activity—subdivided into low gamma (30–50 Hz) and high gamma (50–70 Hz)—to investigate sensory processing and cognitive load [34,36]. The upper bound for high gamma was limited to 70 Hz to minimize artifact contamination.**Autonomic Response—Heart Rate Variability (HRV):** Following the method described in Section 2.4, we analyzed pNN50, an HRV index strongly linked to parasympathetic nervous activity. Higher pNN50 values reflect relaxed states, whereas lower values indicate heightened psychological stress. Accordingly, temporal variations in pNN50 were examined as physiological indicators of emotional change during spatial experiences. We compare the results using this index.**Emotion Mapping in Affective Space:** To enhance interpretability, we constructed a two-dimensional Emotion Map, using pNN50 to represent valence and the β/α ratio to represent arousal. This approach allowed for the visual mapping of emotional states elicited by each spatial condition (see Section 2.4). Additionally, gender-based comparisons were conducted to explore possible differences in emotional responses.**Temporal Analysis and Signal Processing:** To observe time-series changes in autonomic nervous activity between rest and spatial exposure, we generated time-series plots of HRV indicators.

EEG data were collected using OpenBCI at a sampling rate of 200 Hz. Preprocessing included a bandpass filter (1–45 Hz), high-pass filtering (<4 Hz), notch filtering (50 Hz harmonics), low-pass filtering (>47 Hz), and artifact removal via Independent Component Analysis (ICA) using ICLabels. These processes were implemented with MNE-Python [61] and MNE-ICALabel [62,63]. Cleaned EEG signals were segmented by target intervals, and average values were computed. HRV data were derived from R-R intervals provided by MyBeat [60], and pNN50 values were calculated for each analysis interval using Python 3.10. Statistical analyses were conducted using IBM SPSS Statistics (version 28.0.1.1 (14)). The Shapiro–Wilk test was used to assess the normality of each variable. As most physiological indicators (e.g., pNN50 and beta wave power) did not follow a normal distribution, non-parametric Wilcoxon signed-rank tests were used to compare VR and real-world conditions. An exception was made for the change score of the β/α ratio (i.e., stimulus minus baseline), which exhibited normality and was thus analyzed using a paired *t*-test.

### 4.2. Subjective Evaluation of Emotional Responses

As we described in Section 2.3.1, the space evaluation was basically based on intuitive impressions; they are considered suitable for comparing how real and VR spaces influence perception. That is why we used the SD method questionnaire to evaluate the spaces. We show the result of the SD method in Figure 6.

For Place 1, relatively large differences were observed between the VR and real-space conditions in the following adjective pairs: Liked–Disliked, Showy–Plain, Cool–Warm, Inspiring–Ordinary, and Narrow–Spacious. In contrast, adjective pairs such as Calm–Unsettled, Beautiful–Dirty, Tense–Relaxed, Comfortable–Uncomfortable, and Vulgar–Elegant showed differences of less than 0.2 points, indicating minimal variation in impressions. Focusing on the adjective pairs with larger differences, the following trends were found: For Liked–Disliked, the VR space scored 2.583 and the real space 2.119, indicating that the real space was perceived as more liked. For Showy–Plain, the VR space scored 3.982 and the real space 3.143, showing that the real space was perceived as more showy. For Cool–Warm, the VR space scored 4.417 and the real space 5.452, indicating that the VR space was perceived as cooler. For Inspiring–Ordinary, the VR space scored 4.071 and the real space 3.524, with the real space rated as more inspiring. For Narrow–Spacious, the VR space scored 5.185 and the real space 5.690, again suggesting the real space was perceived as more spacious.

These findings suggest that the real space, compared to the VR space, conveyed impressions that were more favorable, spatially expansive, and warmer. In contrast, the VR space tended to evoke impressions that were more neutral or artificial, and it lacked the sensory richness and atmosphere of the real space. This may be attributed to differences in the fidelity of visual information, such as lighting and material textures. In Place 2, relatively large differences between the VR and real spaces were observed for the following adjective pairs: Liked–Disliked, Calm–Unsettled, Beautiful–Dirty, Tense–Relaxed, Comfortable–Uncomfortable, Cool–Warm, and Inspiring–Ordinary. On the other hand, the adjective pairs Showy–Plain, Vulgar–Elegant, and Narrow–Spacious showed score differences of less than 0.2, suggesting minimal differences in impressions.

For the adjective pairs with larger differences, the following patterns were found: Liked–Disliked: VR 3.071, Real 2.143 → Real space was perceived as more liked. Calm–Unsettled: VR 4.190, Real 5.286 → Real space was more calming. Beautiful–Dirty: VR 2.810, Real 2.405 → Real space was perceived as more beautiful. Tense–Relaxed: VR 4.143, Real 4.881 → VR space was perceived as more tense. Comfortable–Uncomfortable: VR 3.214, Real 2.238 → Real space was more comfortable. Cool–Warm: VR 3.381, Real 5.286 → VR space was perceived as cooler. Inspiring–Ordinary: VR 4.667, Real 3.833 → Real space was more inspiring.

These results indicate that the real space tended to be evaluated more positively across multiple dimensions, being described as more liked, calming, beautiful, relaxed, comfortable, warm, and inspiring. In contrast, the VR space was associated with impressions such as tension, coolness, and ordinariness, suggesting it was less emotionally rich in terms of immersion and comfort. This may be due to differences in fine-grained visual information—such as light reflection, spatial depth, and the perception of material textures—that influenced impression formation. The VR space, in contrast, may have provided a more static and uniform visual experience, limiting its ability to convey realism and atmosphere.

When the results from Place 1 and Place 2 are viewed collectively, the real space consistently received higher ratings for emotionally positive impressions such as Liked, Comfortable, and Inspiring. Conversely, for adjective pairs related to arousal, such as Calm–Unsettled and Tense–Relaxed, the VR space was more frequently associated with unsettled or tense impressions. These findings suggest that the VR space may have been perceived as inducing a subjectively higher arousal level compared to the real space.

### 4.3. Physiological Evaluation—Brainwaves

In this section, we evaluate the alpha and beta waves between VR and real-space conditions. As we described in Section 2.3.2, alpha and beta waves are associated with distinct psychological states [48,49]. Alpha waves are typically observed during relaxed or restful conditions, while beta waves are related to attention, cognitive activity, stress, and higher arousal levels. We conducted the statistical comparison for alpha and beta waves measured during the viewing of VR and real spaces at each measurement location. Only data from participants whose recordings were successful were included in the analysis (n = valid samples from 42 participants). All data were preprocessed to remove outliers, as described earlier. Since the data did not follow a normal distribution, the non-parametric Wilcoxon signed-rank test was used for comparison.

Figure 7 shows the results of alpha and beta wave comparisons. For alpha waves, no significant differences were observed between the VR and real-space conditions at either Measurement Place 1 or 2. In contrast, beta wave power was significantly higher in the VR condition at both locations (Place 1: *p* < 0.01, Place 2: *p* < 0.05). These findings indicate that the arousal level of participants was elevated in the VR environment, consistent with previous studies.

Next, to quantify the balance between these states, we computed the β/α ratio, comparing values across the VR and real-world conditions to determine the dominant emotional state. We compared the beta/alpha ratio—a commonly used index of arousal—between conditions (Figure 8). The results showed significantly higher beta/alpha ratios in the VR condition than in the real space at both measurement locations. The beta/alpha ratio indicates a relatively higher power of beta waves compared to alpha waves and is interpreted as a sign of increased arousal.

From these comparisons of alpha waves, beta waves, and the beta/alpha ratio (Figure 7 and Figure 8), we found a consistent pattern of increased beta power and elevated beta/alpha ratios in the VR condition. This suggests that participants experienced heightened arousal and attentional engagement while viewing the VR space. In contrast, alpha wave activity, which is known to increase during meditation, showed no significant difference between the VR and real-space conditions, suggesting that any effect related to meditative states was limited in this context.

In light of these findings, we examined the results of the SD method (Figure 6). Participants rated the VR space more frequently with adjectives associated with higher arousal states, such as unsettled and tense, compared to the real space. These subjective ratings align with the observed increases in beta wave activity and beta/alpha ratios, both of which are physiological indices linked to arousal. Furthermore, this trend is consistent with prior studies reporting elevated arousal during VR use.

Taken together, both subjective and objective results suggest that the VR environment was perceived and experienced as inducing a higher arousal state than the real space. Thus, VR spaces may serve as environments that more readily promote attention and arousal compared to their real-world counterparts.

### 4.4. Memory-Related Brain Activity

As discussed in Section 2.3.2, theta waves are associated with memory encoding and retrieval processes [34,49,50]. In order to examine which of the two environments—virtual reality or real-world space—more effectively stimulates brain activity associated with memory encoding, we conducted a comparative analysis of theta waves as follows.

Figure 9 presents the comparison of theta wave activity during the viewing of VR and real spaces. At both measurement locations, no significant differences were found between the two conditions. Based on this result, it can be inferred that there were no notable differences in memory-related activity—such as encoding and recall—between the VR and real-space conditions.

As described in Section 2.3.2, low gamma waves are associated with the integration of sensory input—such as visual, auditory, and somatosensory stimuli—and with perceptual load and attentional focus in response to visual stimuli [36,37]. In order to determine which environment—VR or real space—induces greater neural activity associated with sensory integration and cognitive processing, we conducted a comparative analysis of gamma waves.

Figure 10 shows the comparison of low gamma and high gamma wave activity when VR and real spaces were presented. For low gamma waves, significantly higher activity was observed in the VR condition compared to the real space at both measurement locations. These results suggest that VR experiences imposed greater perceptual load and required stronger attentional focus than real space. This may be due to the immersive nature of VR and the intensity of visual stimulation, which likely demanded increased multisensory integration.

Regarding high gamma waves, the results indicated a significant increase in VR compared to real space only at Measurement Place 2. No significant difference was found at Place 1. High gamma activity, as explained in Section 2.3.2, is associated with higher-order cognitive processes, integration of visual information, and memory encoding and retrieval [34,35,50].

The increase observed only at Place 2 suggests that the degree of higher-order cognitive engagement—such as memory-related processes—may have differed between Place 1 and Place 2. In particular, the complexity or structural characteristics of the visual stimuli in the VR condition at Place 2 may have prompted greater cognitive processing. Furthermore, although previous studies have reported that interactions between theta and high gamma waves are linked to memory and information processing, no significant differences in theta wave activity were found between VR and real space in the present study.

This suggests that while high gamma activity may indicate an increase in cognitive load, no clear differences were observed between VR and real space in terms of memory-related processes, as reflected by theta wave activity.

### 4.5. Emotion Mapping in Affective Space

#### Autonomic Response—Heart Rate Variability (HRV)

As we explained in Section 2.3.2, the HRV, especially pNN50, evaluated the relaxed states that are generally categorized as positive emotions, and psychologically tense states as negative emotions. Because the distribution of pNN50 values was non-normal, the Wilcoxon signed-rank test—a non-parametric method—was used for analysis. In contrast, β/α values followed a normal distribution, and thus a paired-sample *t*-test was applied.

Figure 11 shows the results of the statistical analysis. The left graph illustrates changes in pNN50. The results show that no significant differences were found between the VR and real-space conditions at either measurement location. Since both environments were physically identical in terms of temperature, humidity, and scent, non-visual sensory input was controlled. This indicates that the differences in neural and emotional responses between VR and real spaces are predominantly driven by visual stimuli, with no significant impact observed at the level of autonomic nervous system activity in this case.

### 4.6. Emotion Mapping in Affective Space

(1)Based on the average evaluation

To enhance interpretability, we constructed a two-dimensional Emotion Map, using pNN50 to represent valence and the β/α ratio to represent arousal after conducting statistical comparisons for the physiological indices beta/alpha (β/α) and HRV (pNN50). With the emotion visualization methodology, which we explained in Section 2.4, it is possible to visualize the effect of both the arousal level and the valence level in the two-dimensional map.

In Figure 12, on the Emotion Map, Place 1 VR1 was plotted in the low position of the first quadrant, Place 1 REAL1 and Place 2 REAL2 in the third quadrant, and Place 2 VR2 in the fourth quadrant. That indicates the distinct emotional responses between the VR and real-space conditions.

Focusing on the Y-axis, which represents arousal, both measurement locations showed that VR conditions were associated with higher arousal than the real space. This trend is consistent across multiple indicators: subjective evaluation results (Figure 6), statistical comparisons (Figure 8), and physiological indices, all of which point to higher arousal in the VR condition.

Regarding the X-axis (valence), both of the VR spaces (VR1, VR2) were plotted in the high valence and middle valence region, while both real spaces (REAL1 and REAL2) were plotted in the third quadrant, corresponding to low arousal and low valence emotions. In accordance with the interviews of the experimental collaborators, we detect one possible reason. That is that the real-space measurement locations were shared environments often used for meetings or meals, and depending on the time of day, these spaces had relatively high foot traffic. As a result, participants may have felt tension or discomfort due to the presence or gaze of others, leading to lower valence scores in the real space.

(2)Focus on the gender-based evaluation

In addition to analyzing overall mean values, we generated multiple Emotion Maps to facilitate comparative analysis. Among these, the gender-based evaluation yielded particularly noteworthy observations. While we initially considered employing inferential statistical methods—such as the Mann–Whitney U test—for gender comparisons, the limited sample size and exploratory nature of the study led us to adopt a descriptive approach instead. Participants were divided into male and female subgroups, and descriptive statistics (mean and variance) were computed for each group. Based on these values, gender-specific Emotion Maps and time-series plots of average physiological responses were constructed to visualize potential differences in emotional experiences.

Figure 13 presents the Emotion Maps averaged separately for male and female participants. As is visually evident, there are notable gender-based differences in emotional responses. Male participants’ responses tended to cluster around the center of the affective space, whereas female responses were more widely dispersed. A quantitative summary of this dispersion is provided in Table 3 and will be discussed in detail later.

Regarding autonomic nervous system activity, as measured by pNN50 along the X-axis (valence), male participants exhibited negligible differences between the VR and real environments in both Place 1 and Place 2. Specifically, the absolute differences were 0.034 for both Place 1 (|0.025 + 0.01|) and Place 2 (|0.014 + 0.02|). In contrast, female participants demonstrated substantially larger variations: 0.19 in Place 1 (|0.10 + 0.07|) and 0.05 in Place 2 (|0.06–0.01|). On average, the female responses were approximately 3.5 times more variable than those of males, suggesting a stronger modulation of autonomic activity in women in response to spatial stimuli.

In terms of arousal (Y-axis, β/α ratio), male participants showed higher arousal in VR1 compared to VR2, while the opposite pattern was observed in real-world settings, where REAL2 elicited higher arousal than REAL1. Among female participants, both VR1 and VR2 conditions resulted in higher arousal than their real-world counterparts, indicating that the VR environments elicited greater neural excitation, particularly among females.

Furthermore, for both genders, alpha wave dominance—typically associated with relaxed or meditative states—was observed in REAL2 (Place 2), suggesting a calming effect of the real environment. Place 2 is characterized by a high proportion of plants in the field of vision, factors which likely contributed to this alpha-dominant, low-arousal state. However, the autonomic response to this space diverged by gender: male participants showed neutral pNN50 values, whereas female participants displayed a marked sympathetic response, indicative of physiological tension.

While gender differences in arousal were minimal in the real environments (Place 1 and Place 2), disparities in autonomic nervous system responses persisted, with female participants generally exhibiting higher sympathetic activation. These findings suggest that female participants were more emotionally sensitive to environmental features and more strongly affected by the immersive and stimulating qualities of VR environments.

Table 3 shows the variance in physiological indices (pNN50 and β/α) calculated separately for male and female participants in the Emotion Map analysis. For pNN50, variance in REAL1 was higher among male participants; however, for VR1, VR2, and REAL2, variance was higher among female participants. This suggests that female participants exhibited greater variability in responses to identical stimuli, particularly regarding valence in VR spaces. For β/α, variance in REAL2 was higher for males, while variance in VR1, REAL1, and VR2 was higher for females. This implies that arousal responses in VR were more varied among female participants. Taken together, these results indicate that female participants showed greater individual differences in both pNN50 and β/α values when viewing VR spaces. This suggests that emotional responses—both in terms of valence and arousal—varied more widely among females. In contrast, male participants exhibited more clustered physiological responses, which may explain the narrower distribution observed in the Emotion Map.

### 4.7. Temporal Analysis and Signal Processing

To observe time-series changes in autonomic nervous activity between rest and spatial exposure, we generated time-series plots of HRV indicators. We can examine the temporal dynamics of physiological responses when viewing VR and real spaces through this analysis.

Figure 14 shows the average pNN50 over time for each stimulus (VR1, REAL1, VR2, REAL2), from resting state through stimulus presentation, based on data from 41 participants with valid pNN50 recordings. Each stimulus and the immediately preceding resting phase are continuous in the time series, although there is a 1–2 min gap between stimulus presentation and the subsequent resting period due to the experimental design.

The results in Figure 14 reveal a temporary increase in pNN50 during the VR stimuli (VR1, VR2), whereas for real-space stimuli (REAL1, REAL2), pNN50 remained mostly flat or showed a slight decline after the stimulus began. As noted in Section 2.3.3, pNN50 reflects parasympathetic nervous activity, with higher values indicating a more relaxed state associated with positive emotional experiences [52,53]. Thus, the observed increase in pNN50 during VR suggests that VR may facilitate parasympathetic activation and relaxation. In contrast, the real-space condition may have induced relatively greater sympathetic dominance and higher arousal.

All participants experienced the spatial environments for the first time during the experiment. Furthermore, in the real-space condition, the experimental areas were public spaces occasionally used for meetings or meals, where participants were exposed to the presence and gaze of others. This social context may have induced psychological tension, contributing to sympathetic dominance. In the VR condition, although ambient sound was still audible, participants could not visually perceive others, which may have reduced social tension and facilitated greater parasympathetic activity.

Figure 15 shows the average pNN50 time series for 28 male participants with valid data. During VR stimuli (VR1, VR2), a gradual increase in pNN50 was observed following stimulus onset, suggesting that male participants exhibited physiological signs of relaxation in response to VR exposure. In contrast, during the real-space stimuli, pNN50 remained stable or declined slightly, indicating weaker parasympathetic activation relative to the VR condition.

Figure 16 presents the pNN50 time series for 13 female participants. Compared to the male participants (Figure 15), pNN50 values were generally higher throughout, suggesting greater baseline parasympathetic activity and a stronger tendency toward relaxation. Among the four stimuli, REAL1 showed the highest peak in pNN50, indicating that this real-space condition was particularly effective in promoting relaxation for female participants.

These findings suggest the possibility of gender-based differences in emotional responses to VR and real environments. Specifically, male participants tended to exhibit higher pNN50 values in VR conditions, whereas female participants showed the highest pNN50 values in real-space conditions. This pattern may indicate that gender-specific characteristics of parasympathetic nervous activity vary depending on the type of environment.

## 5. Discussion

### 5.1. Contributions

**Subjective Evaluation of Emotional Responses**: The results of the Semantic Differential (SD) method revealed notable differences in subjective impressions between the VR and real-space environments. In particular, adjective pairs associated with emotional valence (e.g., Liked–Disliked, Inspiring–Ordinary) tended to receive more favorable ratings in the real space. In contrast, adjective pairs related to arousal (e.g., Calm–Unsettled, Tense–Relaxed) indicated that the VR space elicited a higher level of emotional arousal.

These findings are consistent with physiological data showing increased beta wave activity and higher beta/alpha ratios in the VR condition, objectively suggesting elevated arousal levels (see Section 4.3, Figure 8; Section 4.6, Figure 15). In terms of memory-related brain activity, theta waves—which are associated with memory encoding, recall, and working memory maintenance—did not differ significantly between the two environments (see Section 4.4, Figure 9), suggesting that memory-related responses may not vary substantially between VR and real spaces.

However, an increase in gamma wave activity in the VR condition suggests that VR may impose a higher cognitive and visual processing load. This observation is consistent with previous studies [62], supporting the validity of the present findings. Given the cognitive burden that VR environments may impose on the brain, it may be advisable to limit the duration of exposure when using VR as a substitute for real-space experiences—particularly when the aim is to reduce costs or facilitate pre-implementation evaluations.

Our results suggest that real spaces provide more natural and comfortable experiences. Therefore, improving the fidelity of VR spaces remains a critical challenge if they are to serve as substitutes for real environments.

**Emotion Mapping in Affective Space Based on Physiological Indices**: Emotion mapping based on physiological indices revealed that VR conditions were primarily distributed in the “High Arousal–High Valence” or “High Arousal–Low Valence” quadrants, whereas real-space conditions were predominantly located in the “Low Arousal–Low Valence” quadrant. In the constructed Emotion Map, the X-axis represents valence, as assessed by heart rate variability (HRV) indicators—specifically pNN50—while the Y-axis represents arousal, assessed using EEG-based indices.

As high arousal is physiologically reflected in elevated beta/alpha ratios, both VR conditions (Place 1 and Place 2) exhibited higher arousal levels, consistent with previous studies [25,42]. In contrast, the real-space conditions (REAL1 and REAL2) were characterized by lower arousal and dominant alpha wave activity, as shown in Figure 12 (Section 4.6).

Regarding the valence dimension, which corresponds to parasympathetic activity measured via pNN50, both REAL1 and REAL2 were positioned in the lower-valence (i.e., more tense) region of the map. As illustrated in Figure 12, VR1 and VR2 were associated with higher arousal but not necessarily higher valence. These findings suggest that real spaces may induce stronger autonomic responses and a heightened sense of presence or tension. Therefore, when designing experiences that aim to enhance realism or emotional immersion, real environments may be more suitable than VR alternatives.

**Interpretation of Alpha Wave Activity and Design Intent**: According to Fourcade et al. [48], elevated alpha wave activity is associated with states of relaxation, rest, and creativity. In the present study, both male and female participants exhibited higher alpha wave activity relative to beta waves in the REAL1 condition, which was consistently positioned in the low-arousal quadrant of the Emotion Map.

The designer who created REAL1 reported that the space was intentionally designed to provide a moment of mental reset and to alleviate work-related tension. The observed physiological responses—specifically the dominance of alpha waves—suggest that this design intention was successfully realized.

In addition, both REAL1 and REAL2 showed lower valence scores compared to VR1 and VR2, indicating relatively stronger sympathetic nervous activity in the real-space conditions (see Section 4.6, Figure 12). This heightened physiological tension may be attributed to the real-world experimental context, where other people were physically present during the evaluation. Such social factors may have influenced the participants’ autonomic responses.

To enhance the realism and emotional congruence of VR environments, future developments may benefit from incorporating dynamic elements such as virtual people or social presence cues. These additions could help replicate the emotional impact of real spaces more accurately.

**Gender-Based Differences in Emotional Responses**: A gender-based analysis revealed that female participants exhibited higher arousal levels and greater variability in heart rate variability (HRV) indices in VR conditions compared to their male counterparts (see Section 4.6, Figure 13). This suggests that emotional sensitivity to VR environments may differ by gender.

In contrast, in the real-space conditions, no substantial gender differences in arousal were observed. These findings indicate that gender may play a more prominent role in shaping emotional responses within virtual environments than in real-world settings.

Given these observations, future VR space designs may benefit from considering gender-specific emotional tendencies to optimize user experiences. As research on gender differences in VR-based emotional responses remains limited, this study contributes to the growing body of knowledge in this emerging area.

**Temporal Analysis of pNN50 and Emotional Interpretation**: Time-series analysis of pNN50 revealed a transient increase in parasympathetic nervous activity during VR exposure, indicating a tendency toward relaxation (see Section 4.7, Figure 14). This response may be attributed to the visual isolation provided by the VR environment, which likely reduced social tension by shielding participants from the gaze and presence of others.

In contrast, real-space conditions involved shared environments, where the visibility of others may have induced psychological tension through social presence or perceived observation. These findings suggest that, regardless of visual fidelity, a well-designed VR space can elicit high-arousal emotional states and leave strong impressions on users.

On the other hand, real environments tend to offer more natural and comfortable experiences. Therefore, to position VR as a viable substitute for real-world spaces, further improvements in visual fidelity and environmental realism remain crucial challenges.

Previous studies have not consistently reported such distinctions between VR and real spaces, particularly with respect to emotional and physiological responses. By systematically comparing both subjective evaluations and physiological indices, this study offers new insights that enhance our understanding of the emotional characteristics of VR environments.

**Theoretical and Practical Significance**: From a theoretical perspective, the present study systematically delineates both the similarities and differences in emotional responses to VR and real spaces by integrating subjective evaluations with physiological indices. To our knowledge, large-scale experimental comparisons of this magnitude between real and virtual environments remain scarce; consequently, the practical implications of our findings are substantial.

Future research should investigate how specific spatial elements that constitute VR environments—such as lighting conditions, material textures, and the presence of virtual occupants—affect users’ emotional states and physiological reactions in greater detail.

### 5.2. Limitations and Future Work

One potential limitation of this study lies in the physical and perceptual impact of wearing an HMD. While the use of an HMD may influence emotional or physiological responses independently of the VR content itself, we attempted to minimize this influence by standardizing the conditions as much as possible. Specifically, participants did not wear an HMD during either the resting or stimulus phases in the real-space condition, whereas in the VR space, the HMD was worn throughout both phases. We also considered using the passthrough function of the HMD—allowing participants to view the real space through the device’s built-in cameras—but ultimately decided against it. This was due to concerns regarding the validity of defining the passthrough view as equivalent to direct real-space experience, both in terms of visual fidelity and perceptual realism.

Another limitation concerns the potential influence of color tone differences between the VR and real spaces. In the present study, the color tone of the VR space differed from that of the real environment. Such differences in color characteristics may have influenced emotional responses. For example, a study by Weijs et al. has suggested that different colors within VR environments can elicit different emotional reactions [64]. In future studies, greater attention should be paid to lighting conditions and color rendering in the VR environment to more closely match the visual characteristics of the corresponding real space.

Although the study included 42 participants, which is comparable to prior VR–real comparison studies, the sample size remains relatively limited. This constraint may reduce the statistical power and generalizability of the findings. As such, the results should be interpreted with caution and viewed as preliminary insights. Future studies with larger and more diverse samples, as well as the inclusion of control conditions, are necessary to validate and expand upon these findings.

## 6. Conclusions

This study aimed to evaluate physiological and psychological responses when participants viewed a VR space and a real space with the same spatial design. EEG and heart rate variability (HRV) were used as physiological indices, and subjective impressions were assessed using a questionnaire based on the Semantic Differential (SD) method.

The results indicated that VR spaces tend to elicit higher arousal compared to real spaces. This finding was consistent with both subjective evaluations and physiological indices. Additionally, statistical comparisons of EEG indices revealed increased low gamma wave activity in the VR condition, suggesting a higher load on visual information processing.

Future research will focus on examining how specific elements within VR environments—such as material properties and resolution—affect brainwave indices. Further investigation will also involve time-series analyses and the relationships between multiple EEG indicators.

## Figures and Tables

**Figure 1 sensors-25-04097-f001:**
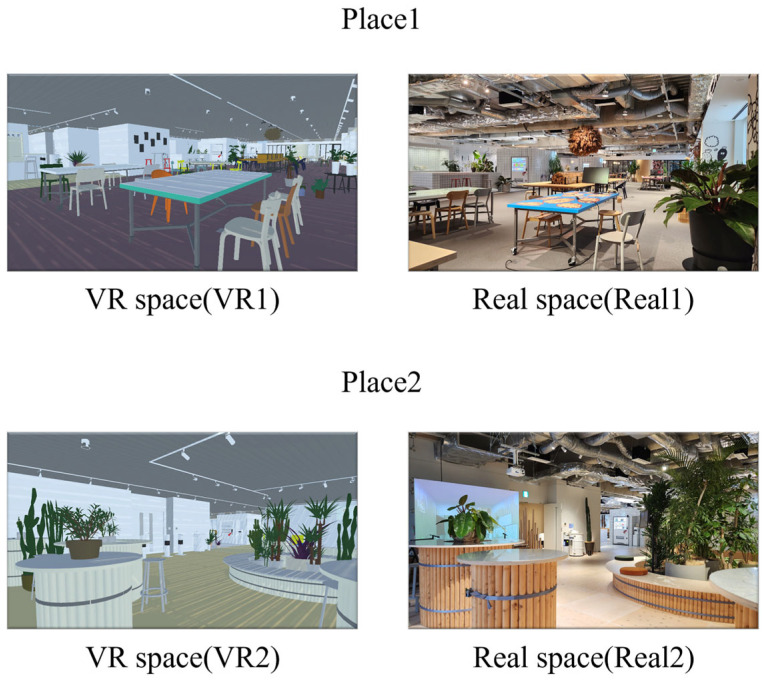
Spaces used in the experiment.

**Figure 2 sensors-25-04097-f002:**
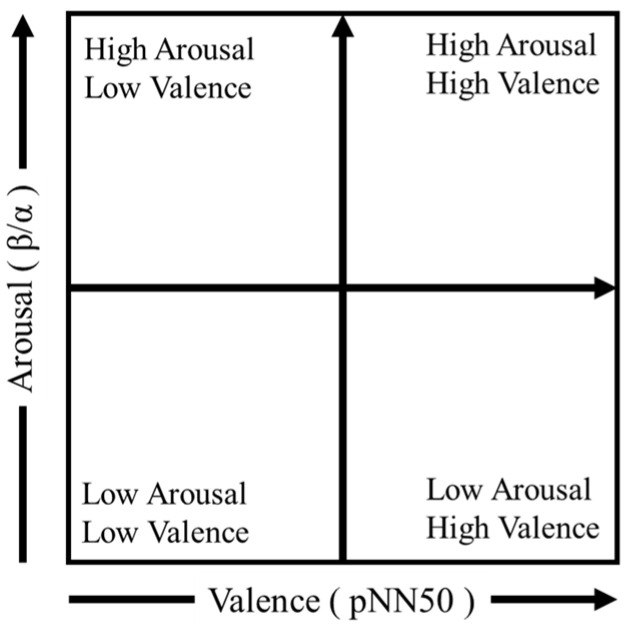
Schematic structure of the Emotion Map used in this study.

**Figure 3 sensors-25-04097-f003:**
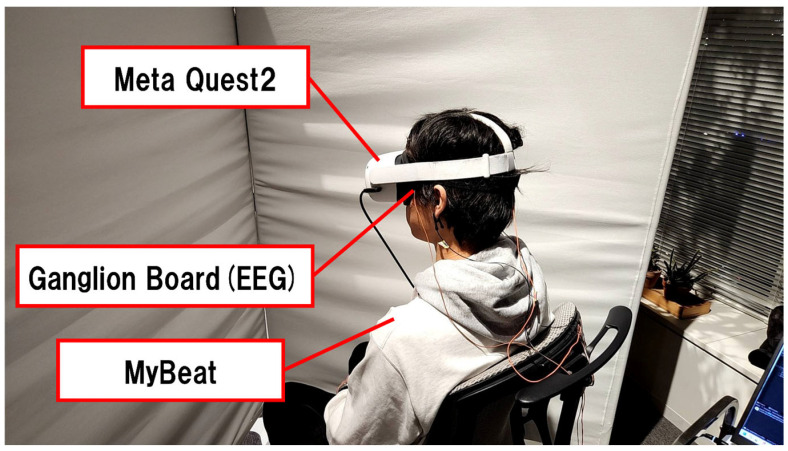
Experimental setup.

**Figure 4 sensors-25-04097-f004:**
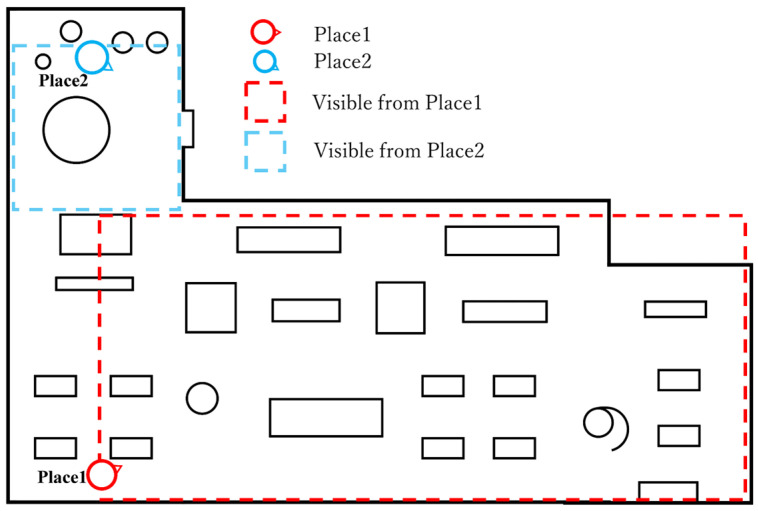
Schematic diagram of the measured spaces.

**Figure 5 sensors-25-04097-f005:**
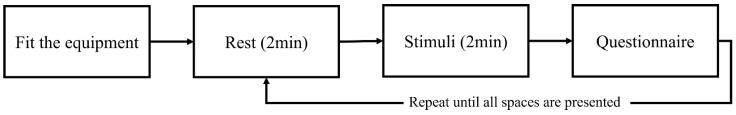
Experimental procedure.

**Figure 6 sensors-25-04097-f006:**
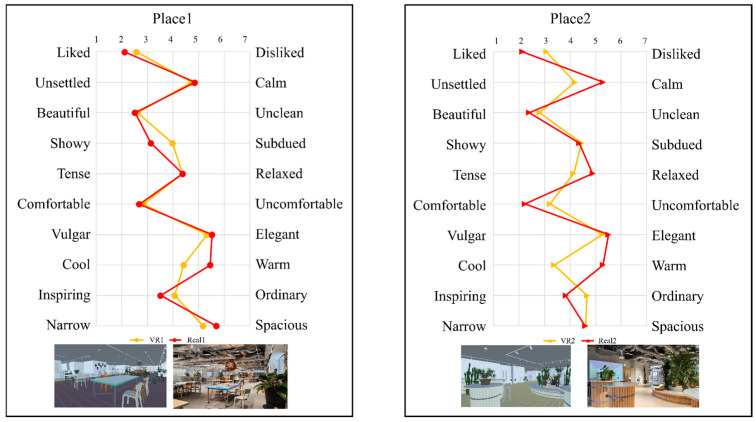
The average ratings from the SD method questionnaire for each space (Place 1 (**left**), Place 2 (**right**)).

**Figure 7 sensors-25-04097-f007:**
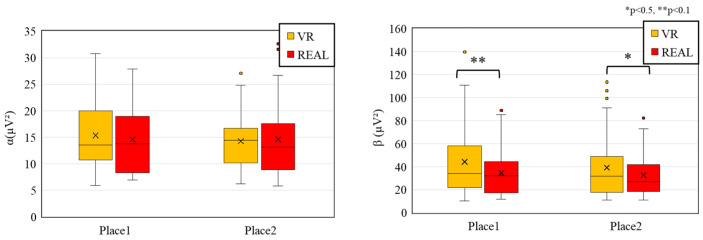
Comparison of alpha and beta waves between VR and real-space conditions.

**Figure 8 sensors-25-04097-f008:**
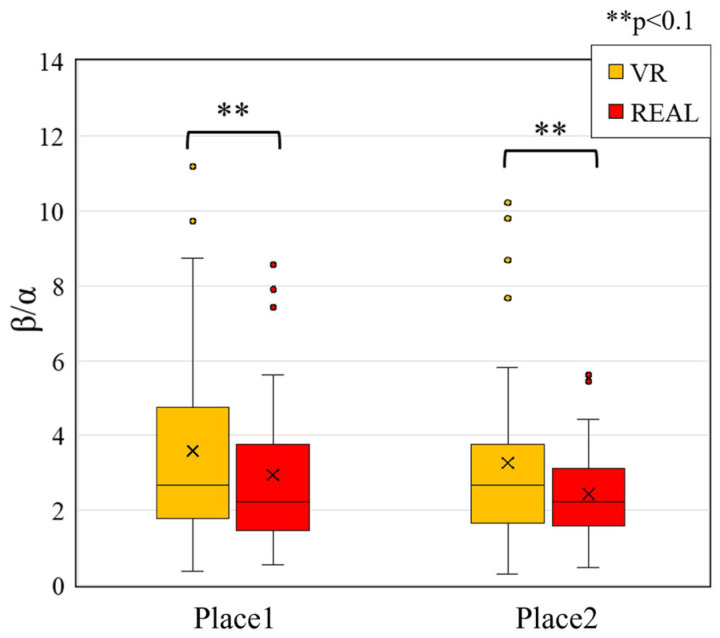
Comparison of beta/alpha ratio between VR and real-space conditions.

**Figure 9 sensors-25-04097-f009:**
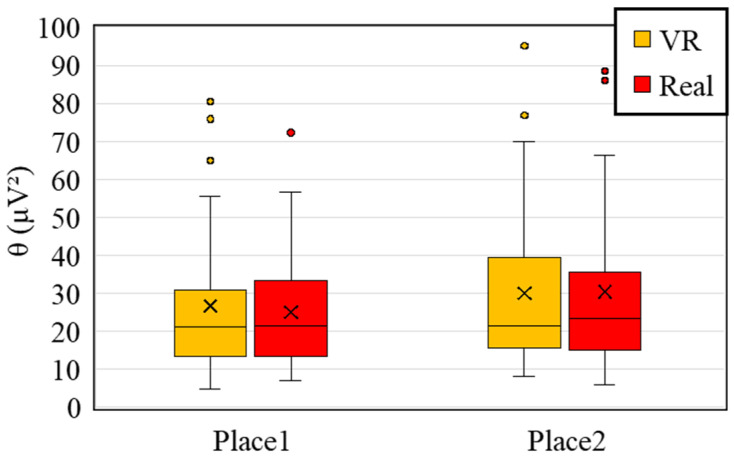
Comparison of theta wave activity between VR and real-space conditions.

**Figure 10 sensors-25-04097-f010:**
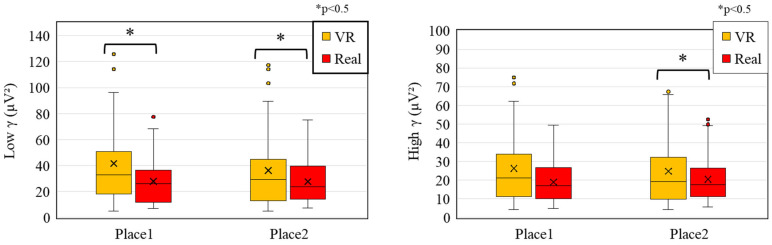
Comparison of low gamma and high gamma wave activity between VR and real-space conditions.

**Figure 11 sensors-25-04097-f011:**
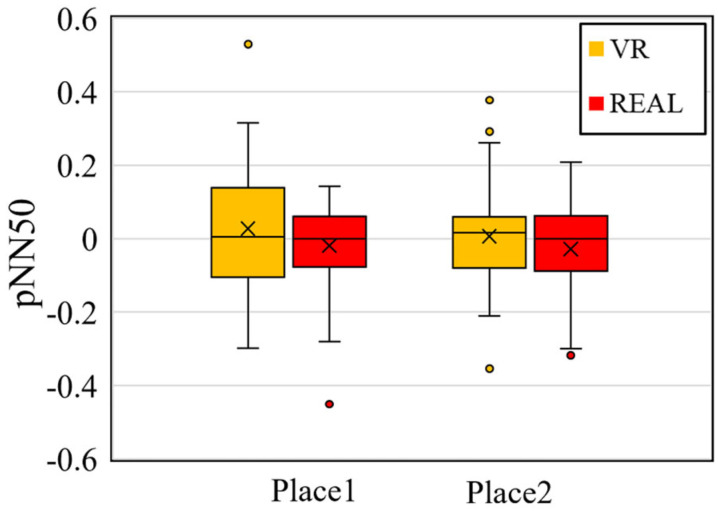
Comparison of autonomic nervous response—heart rate variability, pNN50.

**Figure 12 sensors-25-04097-f012:**
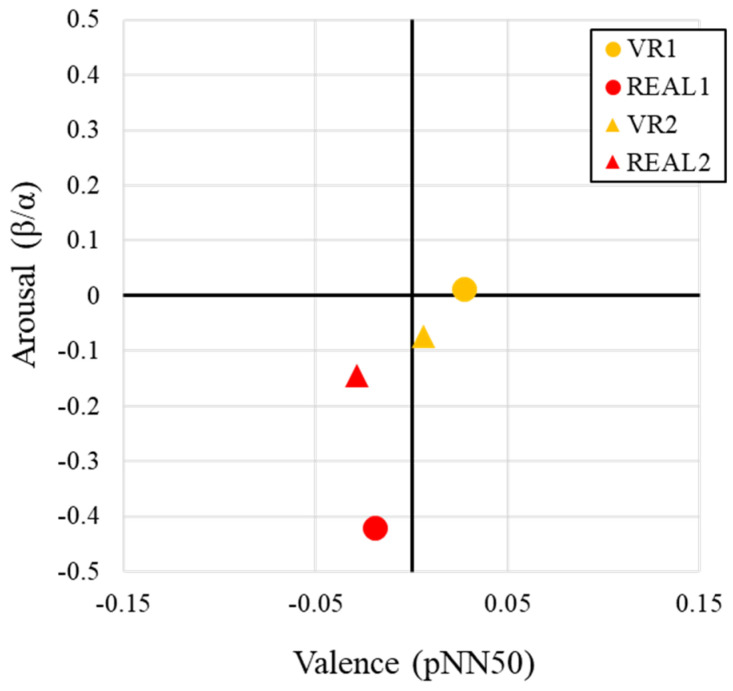
Emotion Map generated from physiological indices during stimulus presentation.

**Figure 13 sensors-25-04097-f013:**
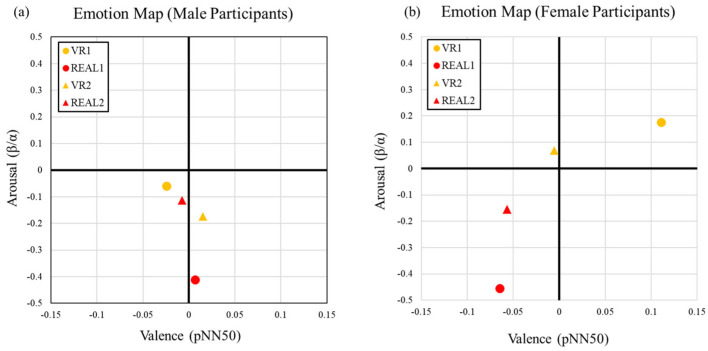
Emotion Map for male participants (**a**) and female participants (**b**).

**Figure 14 sensors-25-04097-f014:**
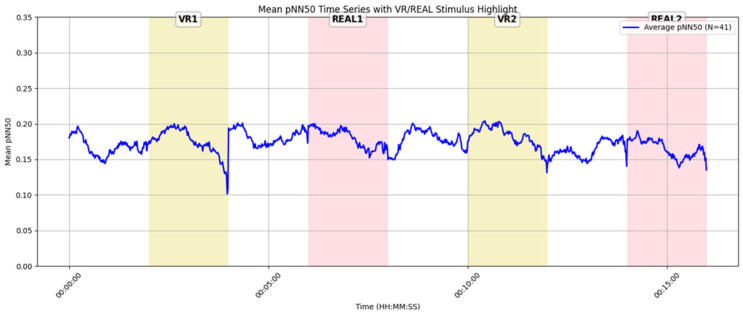
Time-series graph of average pNN50 for all participants (n = 41).

**Figure 15 sensors-25-04097-f015:**
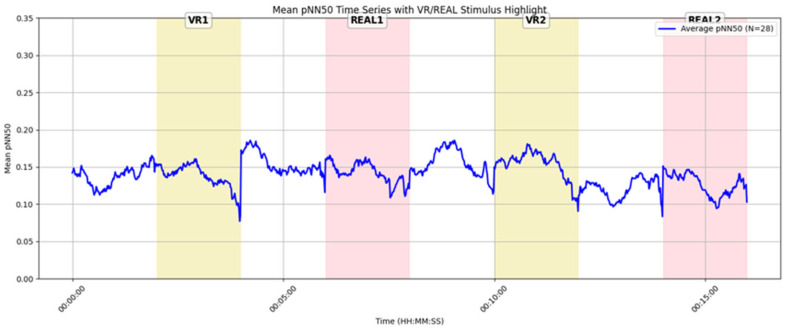
Time-series graph of average pNN50 for male participants (n = 28).

**Figure 16 sensors-25-04097-f016:**
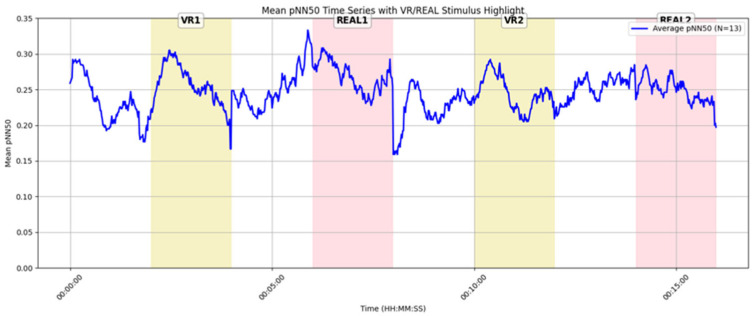
Time-series graph of average pNN50 for female participants (n = 13).

**Table 1 sensors-25-04097-t001:** Adjective pairs used in the SD method.

Adjective Pair	
Liked	Disliked
Calm	Unsettled
Beautiful	Dirty
Showy	Plain
Tense	Relaxed
Comfortable	Uncomfortable
Vulgar	Elegant
Cool	Warm
Inspiring	Ordinary
Narrow	Spacious

**Table 2 sensors-25-04097-t002:** Frequency bands used in EEG analysis.

Wave	Frequency Band (Hz)	Example
Theta	4–7 Hz	Memory encoding and retrieval; working memory
Alpha	8–12 Hz	Relaxation; restful states
Beta	13–30 Hz	Attention; cognition; stress; arousal
Low Gamma	31–50 Hz	Sensory integration; attention; perceptual load; local neural synchronization
High Gamma	51–70 Hz	Higher-order cognitive processes; memory encoding and retrieval; visual information integration; episodic memory

**Table 3 sensors-25-04097-t003:** Variance of pNN50 and β/α for male and female participants.

pNN50	Male	Female	β/α	Male	Female
VR1	0.021356	**0.03918**	VR1	0.562203	**1.12972**
REAL1	**0.01538**	0.007253	REAL1	0.461346	**0.73335**
VR2	0.015968	**0.02675**	VR2	0.822684	**1.11505**
REAL2	0.017831	**0.01891**	REAL2	**0.56983**	0.439284

## Data Availability

The data presented in this study are available on request from the corresponding author. The data are not publicly available due to privacy and ethical restrictions.

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
