# Peer review of "A Pilot Study on Emotional Equivalence Between VR and Real Spaces Using EEG and Heart Rate Variability"

_sensors, 2025, doi:10.3390/s25134097_

Round 1

Reviewer 1 Report

Comments and Suggestions for Authors

Authors proposed the methodology to compare emotional responses in the real space and virtual space (VR technology). This work is interesting and useful.

1) The research methodology is well designed. Data processes are good and designed by considering many factors. However, the authors should rewrite the data processes in order to make readers understand the work easily. Currently, the methods are divided into several small sections so the readers might be unable to capture the entire processes. 

2) In the first half of the manuscript, the explanation is very good as it is easy to understand. In the second half, there is a lot of missing formats, and the explanation of the results is not good (difficult to understand). 

3) In the results, the effects of VR on the emotional responses are significant. I am just wonder that the difference is the effects of wearing the VR glasses not from the visual using VR technology. In addition, the color tone in VR is different from that in the real space (VR = blue, Real space = yellow). Does it affect the results?

4) Is it possible that the number of participants is not enough? In my opinion, for this kind of experiments, the number of participants should be greater than 100 participants in order to avoiding bias in the results.

5) Authors should indicate whether the VR use is suitable or not (or could not decide yet) to replace the access of real space (especially for the current work). What kind of the application is the VR technology suitable with a little bias? 

Author Response

Please see the uploaded Response Letter (Word file) below for our detailed replies to all reviewer comments, including yours. Thank you very much for your valuable feedback.

Reviewer 2 Report

Comments and Suggestions for Authors

This is a wonderful and methodologically strong study that addresses an innovative and timely topic. The combination of VR environments with psychophysiological measures such as EEG and HRV offers a promising avenue for understanding emotional responses. The manuscript is engaging and offers valuable insights. Nevertheless, there are several areas where clarity, structure, and methodological transparency could be improved.

  1. Given the exploratory nature of the research, the relatively small sample size, and the absence of a control group, I strongly suggest that the authors explicitly frame this as a pilot study. This should be reflected in the title, abstract, and at relevant points throughout the manuscript to appropriately set expectations and highlight the study's preliminary nature.

  2. There is a significant portion of the introduction (lines 62–77) without any supporting references. Since this section includes background information and conceptual framing, please ensure proper citations are included to support the claims and contextualize the study.

  3. The aims of the study are currently embedded within the methods section. For improved clarity and readability, I recommend presenting a dedicated subsection in the introduction clearly outlining the study’s objectives and hypotheses (if any). This helps orient the reader earlier in the manuscript.

  4. It would be beneficial to include an image/visual representation of the emotional map used by participants. This would enhance reader comprehension and add transparency to the tools used in the data collection process.

  5. While the methods provide a solid overview of the physiological measures (EEG and HRV), there is insufficient detail about the data analysis procedures. Specifically:

    • Which statistical software was used?

    • What statistical tests were applied (e.g., t-tests, medians, ANOVA)?

    • Was the normality of data assessed consistently across all variables and analyses, as done in section 3.5.2?

    • These procedures should be detailed in a dedicated "Statistical or Data Analysis" subsection within the Methods section, rather than primarily reported in the Results. A brief summary can remain in the Results to aid navigation, but the main explanation belongs in the methodology.

    • For instance, how were metrics like pNN50 variance and the β/α ratio calculated for male and female participants? These computational details are essential for reproducibility and should be explained in the methods.

  6. Figure 10, referenced at lines 521 and 532, appears to be missing from the manuscript. Please ensure this figure is included and properly labeled.

  7. In analyzing differences by gender, did you consider using other statistical techniques? This would strengthen the validity of gender-based comparisons.

  8. The discussion section would benefit from further elaboration. In particular:

    • Results should be interpreted, not just reported.

    • A comparison with existing literature is needed to situate your findings within the broader research landscape.

    • Add a subsection specifically addressing implications and future research directions, as this would increase the paper's theoretical and practical relevance.

  9. Currently, the paper does not contain a dedicated section on the limitations and strengths of the study. Including this would help contextualize the findings, especially given the pilot nature and sample size. Consider addressing aspects such as ecological validity, potential biases, and generalizability.

Author Response

(The authors gave the same response as above.)

Round 2

Reviewer 1 Report

Comments and Suggestions for Authors

The authors' explanation is excellent. The manuscript is acceptable for publication.

Reviewer 2 Report

Comments and Suggestions for Authors

The authors have thoroughly responded to all the comments I provided and enriched the manuscript with additional data, including noteworthy insights into gender differences.